# The Six-Transmembrane Enzyme GDE2 Is Required for the Release of Molecularly Distinct Small Extracellular Vesicles from Neurons

**DOI:** 10.3390/cells13171414

**Published:** 2024-08-24

**Authors:** Kyle T. Shuler, Josue Llamas-Rodriguez, Reuben Levy-Myers, Shanthini Sockanathan

**Affiliations:** 1The Solomon Snyder Department of Neuroscience, The Johns Hopkins School of Medicine, 725 N Wolfe Street, Baltimore, MD 21205, USA; kshuler1@jh.edu (K.T.S.); jllamas3@jh.edu (J.L.-R.); rlevymyers@health.ucsd.edu (R.L.-M.); 2Department of Neurosciences, University of California, San Diego, 9500 Gilman Drive, La Jolla, CA 90293, USA

**Keywords:** extracellular vesicles, ectosomes, neurons, GDE2, synapse, cytoskeleton, redox

## Abstract

Extracellular vesicles (EVs) are implicated in a multitude of physiological and pathophysiological processes in the nervous system; however, their biogenesis and cargoes are not well defined. Glycerophosphodiester Phosphodiesterase 2 (GDE2 or GDPD5) is a six-transmembrane protein that cleaves the Glycosylphosphatidylinositol (GPI)-anchor that tethers some proteins to the membrane and has important roles in neurodevelopment and disease-relevant pathways of neuronal survival. We show here that GDE2 regulates the number of small EVs (sEVs) released from the cell surface of neurons via its GPI-anchor cleavage activity and contributes to the loading of protein cargo through enzymatic and non-enzymatic mechanisms. Proteomic profiling reveals that GDE2 releases at least two distinct EV populations, one containing GDE2 itself and the other harboring the putative ectosomal markers CD9 and BSG. sEVs released by GDE2 are enriched in cytoskeletal and actin-remodeling proteins, suggesting a potential mechanism for GDE2-dependent EV release. Further, sEV populations released by GDE2 are enriched in proteins responsible for modulating synaptic activity and proteins that are critical for cellular redox homeostasis. These studies identify GDE2 as a novel regulator of molecularly distinct sEV populations from neurons with potential roles in the synaptic and redox pathways required for neuronal function and survival.

## 1. Introduction

EVs are membranous particles released by bacteria, plants, and animals that carry nucleic acids, lipids, and proteins. Initially thought to have roles limited to the disposing of cellular waste, EVs are now appreciated as purveyors of information between cells [1,2,3]. In the central nervous system (CNS), EVs are implicated in various physiological processes, including the maintenance of myelination, synaptic plasticity, trophic support, and penetrance across the blood–brain barrier [4,5,6]. More recently, EVs have emerged as a possible mechanism that underlies the stereotypical progression of pathologies across brain regions in neurodegenerative diseases such as Alzheimer’s disease (AD) and Amyotrophic Lateral Sclerosis (ALS) [7,8]. However, EVs also exhibit neuroprotective properties; for example, EVs are implicated in the transfer of molecular cargoes that suppress protein aggregation in other cells [9,10]. How EVs implement these contrasting toxic and neuroprotective functions is unclear. Deeper insight into the types of EVs released in the nervous system and their biogenesis and cargoes is needed to better understand the complexity of their roles in physiology and disease.

EVs are highly heterogeneous and can be classified as exosomes or ectosomes (also known as microvesicles), according to their mechanism of biogenesis [11,12,13]. Exosomes are generally <200 nm in diameter and formed via the endosomal pathway, which generates multivesicular bodies (MVBs) that release exosomes into the extracellular space upon fusion with the cell surface [11,13,14]. Exosomal cargo loading is mainly regulated by ESCRT (endosomal sorting complex required for transport) proteins, tetraspanins, and ceramides during the process of MVB formation, and the exosomes are subsequently released via the recruitment of Rab small GTPases and SNARE proteins [11,15]. Ectosomes, in comparison, are formed via direct outward budding of the plasma membrane into the extracellular space and carry cargo that reflects their mode of biogenesis (i.e., cytoplasmic proteins and plasma membrane lipids) [14,16]. Emerging studies suggest that exosomes and ectosomes may each be composed of molecularly distinct subtypes released by designated pathways [14,16,17], allowing the fine-tuned release of specific EV subtypes in response to stimuli while maintaining global EV homeostasis. EVs can also be classified according to their size as small (sEV, <200 nm) or large (lEV, >200 nm), which does not specify a mechanism of biogenesis [13,18]. As the majority of EV preparations consist of overlapping subtypes, the more general term of EV is typically most appropriate, unless a particular size range or mechanism of biogenesis can be demonstrated [13,19].

Glycerophosphodiester phosphodiesterase 2 (GDE2 or GDPD5) and its family members, GDE3 and GDE6, are six-transmembrane proteins that operate at the cell surface to cleave the glycosylphosphatidylinositol (GPI)-anchor that secures specific proteins to the cell membrane [20,21,22,23,24,25,26]. GDE2 is primarily expressed in neurons but is also detected in terminally differentiated oligodendrocytes and vascular endothelial cells [27], while GDE3 is found in astrocytes and oligodendrocyte precursor cells (OPCs) [20,28]. GDE6 expression in the mammalian nervous system has not been detected [29]. Developmentally, GDE2 promotes motor neuron differentiation by downregulating Notch signaling through the inactivation of the REversion-inducing Cysteine-rich protein with Kazal motifs (RECK) via cleavage of its GPI-anchor [25]. GDE2 has separate roles in the adult nervous system, where it is required for neuronal survival [27,30]. A timed ablation of GDE2 that leaves its embryonic function intact results in motor neuron and cortical neuronal loss, neurodegenerative changes, and cortical abnormalities such as TDP-43 mislocalization, the reduction of synapses, synaptic protein loss, and the accumulation of toxic Aβ42 peptides [23,27,30]. The failure of the GDE2 cleavage of RECK is linked to synaptic deficits and an increase in Aβ42, while TDP-43 abnormalities arise from the aberrant activation of canonical Wnt signaling in neurons as a result of GDE2 dysfunction [23,30]. The mechanism by which the loss of GDE2 stimulates neuronal Wnt activation remains unclear but is hypothesized to occur through the GPI-anchor cleavage of relevant substrates. The cellular abnormalities observed in mice lacking GDE2 (*Gde2*KOs) reflect the neuropathologies observed in AD, ALS, and ALS/frontal temporal dementia (FTD), and *Gde2*KO mice show motor and cognitive changes observed in mouse models of AD, ALS, and ALS/FTD [31,32]. An analysis of postmortem tissue shows that GDE2 forms aberrant intracellular accumulations in the brain of patients with AD and ALS, and consistent with GDE2 dysfunction in disease, the amounts of membrane-tethered and soluble RECK are respectively increased and decreased in AD brain, and the amounts of GPI-anchored proteins are disproportionately reduced in the cerebrospinal fluid of patients with ALS [23,31]. TThus, the GPI-anchor cleavage activity of GDE2 is a central feature of its function in the developing and adult nervous system, and disruptions in GDE2 function in the adult may contribute to neuropathologies in AD, ALS and ALS/FTD.

Recent studies of the related protein GDE3 reveal that GDE3 is capable of releasing EVs in glial cells [20,28]. GDE3 controls OPC proliferation by stimulating ciliary neurotrophic factor (CNTF) signaling via the bimodal release of the CNTF receptor-α subunit through EV release and GPI-anchor cleavage [20]. In astrocytes, GDE3 utilizes mechanisms independent of its enzymatic function to regulate actin dynamics via the WAVE complex, to release a molecularly distinct EV subtype from the plasma membrane that acts on neurons to modulate postsynaptic activity [28]. Thus, at least in astrocytes, GDE3 feeds into major EV release pathways to fine-tune the release of specific EV subtypes important for inter-cellular communication. Whether GDE2 is involved in the release of EVs from neurons and if this function underlies its roles in the developing and adult nervous system is not known. Here, we use gain- and loss-of-function approaches across mouse and human cellular models to investigate the influence of GDE2 on EV release. We find that GDE2 is necessary and sufficient to promote the release of neuronal EVs via its GPI-anchor cleavage function and to contribute to the loading of select EV cargoes through enzymatic and non-enzymatic mechanisms. Studies in HEK293T cells and primary neurons are consistent with the model that EVs released by GDE2 are predominantly sEVs and that these EVs can be stratified into two broad classes according to the presence of GDE2 itself. Further, proteomic analyses of established EV markers indicate that both classes of EVs contain SDCBP and TSG101, while EVs lacking GDE2 uniquely contain the putative ectosomal markers CD9 and BSG. Notably, Gene Ontology (GO) analyses revealed that both classes of sEV released by GDE2 shared cellular compartment and molecular function terms pertaining to the synapse and redox function. Taken together, our studies identify GDE2 as a novel regulator of sEV release from neurons and propose potential functions of these EVs in synaptic and neuronal redox biology that may be relevant to the roles of GDE2 in neurodifferentiation and survival in developing and adult nervous systems. These studies highlight the six-transmembrane GDEs as important regulators of EV release in the nervous system.

## 2. Materials and Methods

### 2.1. Animals

Animals were maintained and used in accordance with approved Johns Hopkins University IACUC protocols. *Gde2* WT (*Gde2^+/+^*), KO (*Gde2^−/−^*), and heterozygous (*Gde2^+/−^*) mice were bred, maintained, and genotyped as described previously [27]. Mouse lines expressing the GDE2 protein with an external 3X FLAG tag between K423 and G424 were generated by the Johns Hopkins Transgenic Mouse Core. One-cell B6SJLF2/J embryos (Jackson Laboratories, 100012, Bar Harbor, ME, USA) were generated using protocols as described [33]. Embryo electroporation was performed using a protocol adapted from Kaneko et al. [34]. The CRISPR RNA (crRNA) and oligo sequence are shown below. An electroporation solution was made in a stepwise progression, starting with equal volumes of Cas9 protein (1000 ng/µL stock, IDT), tracrRNA (20 µM stock, IDT), and crRNA (20 µM stock, IDT) combined to make a 9 µL RNP solution. RNAse-free injection buffer (10 mM Tris-HCl, pH 7.4, 0.25 mM EDTA) was used to prepare all stocks and incubated on ice for 10 min. An RNP + oligo solution was then made by adding 1 µL of DNA oligo (2000 ng/µL stock, IDT) to the RNP solution and moving it to room temperature. A final electroporation mix was made by adding 35 µL Opti-MEM (Gibco, 31985062, Waltham, MA, USA) to the RNP + oligo solution at room temperature. Embryos were placed in a 5 mm gap slide containing 45 µL electroporation solution in groups of up to 100. The following parameters were used for electroporation: poring pulse (voltage: 225 V; pulse length: 1.0 ms; pulse interval: 100 ms; number of pulses: 4; decay rate: 10%; polarity: +), transfer pulse (voltage: 20 V; pulse length: 50 ms; pulse interval: 50 ms; number of pulses: 5; decay rate: 40%; polarity: ±). Following electroporation, the embryos were washed in Advanced KSOM (Sigma, MR-101-D, Burlington, MA, USA) and moved to a 5% CO_2_ incubator. Two hours after electroporation, the embryos were transferred into pseudopregnant CD1 females (Charles River, Wilmington, MA, USA) using protocols described in Nagy et al., 2003 [33].

crRNA: GCCATCGCTAACTTACGGAA

Oligo: GAAGATGGCTCCTGGCTTCCAGCAAACATCTGGATCCAAAGAAGCCATCGCTAACTTAAGAAAAGGGGATTACAAGGATGACGACGATAAGGACTATAAGGACGATGATGACAAGGACTACAAAGATGATGACGATAAAGGTCACATCCAGAAGCTGAACCTCCGCTACACTCAGGTGTCCCACCAGGAGCTCAGGTGCC

### 2.2. Antibodies

All antibodies used in this study are listed here, and concentrations used for specific experiments are listed in the relevant subsections: α-Alix (Cell Signaling Technologies, 2171S, Danvers, MA, USA), α-annexin-a1 (Abcam, ab214486, Waltham, MA, USA), PE-conjugated α-CD63 (BD Biosciences, 564222, 557305, Franklin Lakes, NJ, USA), PE-conjugated α-CD81 (BD Biosciences, 559519, 555676, Franklin Lakes, NJ, USA), biotinylated α-FLAG (Cell Signaling Technologies, 2908, Danvers, MA, USA). α-FLAG primary antibody (Cell Signaling Technologies, 14793S, Danvers, MA, USA), α-FLAG (Millipore, F1804, Burlington, MA, USA), PE-conjugated α-FLAG (OriGene, OTI4C5, Rockville, Maryland, USA), α-NCAM-1 (R&D Systems, AF2408, Minneapolis, MN), PE-conjugated α-NCAM-1 (BD Biosciences, 563238, 753644, Franklin Lakes, NJ, USA), α-NeuN (Synaptic Systems, 266 004, Goettingen, Germany), and α-RECK (Cell Signaling Technologies, 3433, Danvers, MA, USA).

### 2.3. Tissue Dissection and Immunohistochemistry

Brain tissue from GDE2-eFLAG and WT mice were embedded in Tissue-Tek^®^ O.C.T. compound and stored at −80 °C. The tissue was sectioned into 20 µm thick sections using a CM3050 S Cryostat ( Leica, Wetzlar, Germany) with a working temperature of −20 °C. Sections were washed 3 times for 5 min in 1X PBS and incubated in 5% Bovine Serum Albumin (Millipore Sigma, Burlington, MA, USA) in 1X PBS for 3 h at room temperature. Sections were incubated overnight at 4 °C with 1:500 α-FLAG primary antibody (Cell Signaling Technologies, 14793S, Danvers, MA, USA) diluted at a 1:500 concentration in 1% Bovine Serum Albumin in 1X PBS. Sections were then permeabilized with Triton-X and incubated with α-NeuN primary antibody at 1:3000 in 1% BSA in 1X PBS overnight at 4 °C. Sections were then washed 3 times for 5 min in 1X PBS. Corresponding secondary antibodies (FITC Guinea Pig for α-NeuN and Cy3 rabbit for α-FLAG) were diluted at 1:500 concentration and incubated with the slides for 1 h at room temperature in the dark. Hoescht dye was diluted at a 1:1000 concentration and incubated with the tissue alongside the secondary antibodies. Slides were washed 5 times for 5 min in 1X PBS and mounted with ProlongGold. Negative control sections were processed along with the same protocol, but the primary antibodies were omitted.

### 2.4. Cell Culture Staining

Cultured cells on coated coverslips were washed in PBS and fixed in 4% PFA in phosphate buffer for 15 min on ice. Cells were then washed in PBS three times for 5 min. After washing, the coverslips were blocked in PBST containing 5% BSA (Bovine Serum Albumin) for 1 h. Primary antibodies were diluted in 1% BSA in PBST (at the concentrations indicated below) and incubated overnight at 4 °C. The next day, the slides or coverslips were washed in PBST three times for 5 min. Appropriate secondary antibodies at 1:500 and Hoechst at 2 mM were diluted in PBS and incubated with coverslips for 1 h at room temperature in the dark. α-FLAG-M2 (Millipore, Burlington, MA, USA) primary antibody was utilized at a 1:500 dilution. MemGlow488 (Denver, CO, USA) was then utilized at a concentration of 100 nM for 10 min at RT to label the cell membranes. Coverslips were then washed in PBST 3 times for 5 min, and then mounted with ProlongGold (Thermo Fisher, Waltham, MA, USA). Cells on the coverslips were imaged on a LSM 700 (Zeiss, Oberkochen, Germany) confocal microscope.

### 2.5. Plasmids and Cloning

Expression plasmids containing GDE2 and RECK are described in Park et al., 2013 [25]. The GDE2-H243A construct was generated using site-directed mutagenesis using the primer 5′ctcccatgctggctccagaaGCCacagtgatgtccttccggaaggcgctggagcagaggc. The mouse GDE2-eFLAG construct was generated using Gibson cloning to insert a 3X FLAG tag at K423 using the primers

5′ tgacgatgataaggattataaggacgacgatgacaagGGTCACATCCAGAAGCTG and

3′ tctttgtaatctttatcgtcgtcgtccttgtagtccccTTTCCGTAAGTTAGCGATG. The position for insertion of the tag was selected with guidance from the 3D structure predicted by AlphaFold [35].

### 2.6. HEK293T Cell Culture and Transfection

HEK293T cells were grown in HEK medium [DMEM (Dulbecco’s modified Eagle’s medium) (Gibco, 31053028, Waltham, MA, USA), 10% FBS (Millipore Sigma, F4135-500, Burlington, MA, USA), and 1% PenStrep (Gibco, 15140122, Waltham, MA, USA) in a humidified 37 °C incubator with 5% CO_2_. The plates were coated with PEI (polyethylenimine; 25 μg/mL) (Sigma-Aldrich, P3143, Burlington, MA, USA) for 1 h and washed with PBS three times before use. When confluent, HEK293T cells were washed in PBS and trypsinized with TrypLE (Gibco, 12563011, Waltham, MA, USA) for 5 min. HEK293T cells were transfected the day after plating with Lipofectamine 2000 (Thermo Fisher Scientific, 11668027, Waltham, MA, USA) according to the manufacturer’s protocol. Briefly, 405 µL of OptiMEM (Gibco, 31985070, Waltham, MA, USA) was mixed with 24.3 µL of Lipofectamine 2000 and incubated at room temperature for 5 min, then mixed with 405 µL of OptiMEM containing 100 ng of DNA and allowed to incubate for 30 min at room temperature. This mixture was applied to HEK293T cells at a 40 to 50% confluency in a 10 cm dish. This process was scaled as necessary depending on the number of groups transfected. The transfected cells were then utilized for experiments as detailed in the following sections.

### 2.7. Primary Cortical Neuron Isolation and Culture

Cells from post-natal day 0 or 1 mouse cortices were seeded at 1.5 × 10^5^ cells/cm^2^ on poly-L-lysine (PLL, Sigma Aldrich, P2636-100MG, Burlington, MA, USA)-coated T25 flasks and initially plated in Neurobasal medium (Thermo Fisher, 21103049, Waltham, MA, USA) supplemented with 10% FBS, 2% glucose solution (20% *w*/*v* Thermo Fisher, A2494001, Waltham, MA, USA), 1% sodium pyruvate (Gibco, 11360070, Waltham, MA, USA), and 1% PenStrep for 1 h. After this incubation period, the medium was switched to Maintenance Medium consisting of Neurobasal medium supplemented with 2% B27-Plus (Gibco, A3582801), 1% GlutaMAX (Thermo Fisher, 35050061, Waltham, MA, USA), and 1% PenStrep. Cytosine arabinoside (AraC, Sigma Aldrich, Burlington, MA, USA) was added on day in vitro (DIV) 3 to inhibit glial proliferation. From DIV4 onwards, the cultures were fed every 3 days via a half-medium change with Maintenance Medium. Cultures were maintained at 37 °C until harvesting at DIV14–17 following media harvest.

### 2.8. Extracellular Vesicle Purification and Characterization

EV purification and characterization were performed in adherence to the guidelines detailed in the minimal information for studies of extracellular vesicles (MISEV2023) [13]. For HEK293T cultures, 3.5 × 10^6^ cells were utilized per group for EV production. Following transfection, the culture medium was switched to fresh HEK293T medium for 24 h. Following this period, the cells were washed twice with 1X phosphate-buffered saline (PBS), and the medium was then switched to serum-free DMEM with 1% p/s for 24 h for EV production. There was no observable effect of the transfection on cell viability or morphology. For neuronal cultures, 3.75 × 10^6^ cells were utilized and the conditioned Maintenance Medium from DIV14–17 was harvested on DIV17. There were no observable differences in cell viability or morphology between the WT and *Gde2*KO cultures. The EV-conditioned medium was initially clarified of cell debris utilizing serial centrifugation at 300× *g* for 10 min, followed by an additional round at 2000× *g* for 20 min. Following clarification, the conditioned medium was concentrated to 1 mL and passed through a qEV1/35 nm size exclusion chromatography column (IZON, Christchurch, New Zealand). Following elution of the buffer volume, 5 fractions (3.5 mL total) were collected as the EV-enriched fractions per the manufacturer’s instructions. The EV fractions then underwent ultracentrifugation at 100,000× *g* for 70 min using a Beckman Optima TLX Ultracentrifuge with TLA100.4 rotor (Beckman Coulter, Brea, CA, USA). Depending on the downstream applications, EV pellets were resuspended in 100 µL of either 1X PBS or lysis buffer (see Western blot section). Following purification, the EV preparations were characterized utilizing Western blotting to analyze the EV markers and nano-flow cytometry (nFC) to obtain a particle count, size, and orthogonal measurement of EV protein markers, as well as transmission electron microscopy (TEM) for a qualitative assessment of the EV preparations and an orthogonal measurement of the EV size. Specific experimental details for each of these methodologies are included in the following sections.

### 2.9. Western Blot

Samples in gel loading buffer (GLB) were boiled for 5 min. BME was added to a final concentration of 0.1%. Fifteen microliters of samples and 5 µL of protein ladder was loaded on a 10% Criterion™ TGX™ Stain-Free™ Precast Midi Protein Gel (BioRad, 5678035, Hercules, California, USA) and run with tris-glycine buffer (25 mM tris, 192 mM glycine, and 1% SDS) at 200 V for 45 min. Gels were UV-activated using the ChemiDoc MP Imaging System (BioRad, 12003154, Hercules, California, USA). Gels were transferred to a methanol-soaked PVDF (polyvinylidene fluoride) (Millipore, IPVH00010, Burlington, MA, USA) membrane for the use of BioRad Trans-Blot Turbo set (Hercules, California, USA)to 25 V/1.0 A for 30 min at RT. Membranes were blocked in EveryBlot blocking buffer (BioRad, 12010020, Hercules, California, USA) for 1 h at RT with gentle shaking. Membranes were incubated in primary antibodies (at the concentrations indicated below) in EveryBlot blocking buffer overnight at 4 °C with gentle shaking. The following day, the membranes were washed 6 times for 5 min in 0.3% TBST. Appropriate secondary antibodies (Kindle Biosciences, R1005 or R1006, Greenwich, CT, USA) at 1:1000 in EveryBlot blocking buffer were applied to the membranes for 1 h at room temperature with gentle shaking. Membranes were then washed 6 times for 5 min in 0.3% TBST. ECL (enhanced chemiluminescence) (Kindle Biosciences, R1002, Greenwich, CT, USA) substrate was applied to the membranes for 4 min, and the membranes were immediately imaged using the ChemiDoc MP Imaging System (BioRad, Hercules, CA, USA. The primary antibodies are the following: α-alix (1:2000), α–annexin A1 (1:1000), α-NCAM-1 (1:1000), α-RECK (1:1000), α-FLAG (1:10,000). Western blots were quantified using Fiji [36]. We analyzed EVs derived from GDE2-transfected HEK293T cells as compared to control cells that were either transfected with H243A mutant GDE2 or received no transfection (NT). Utilizing this strategy, we evaluated the effects of GDE2 expression, and the involvement of its catalytic domain, on the expression of the EV markers Annexin A1, Alix, and NCAM-1 by calculating the fold changes in the mean chemiluminescent intensity between the GDE2 and H243A groups and NT group. The mean chemiluminescent intensity was calculated for each marker and was divided by the mean chemiluminescent intensity value calculated from the corresponding cell lysate sample. As mentioned in the previous section, 3.5 × 10^6^ cells were utilized across each experimental group with no observable effects of transfection on cell viability or morphology. Three biological replicates were utilized for each experimental group.

### 2.10. Nano-Flow Cytometry

For nano-flow cytometry data acquisition, a NanoFCM Nanoanalyzer (Nottingham, England) in the Johns Hopkins EXCEL—EXtracellular particle Characterization and Enrichment Lab—was utilized. The nano-analyzer was used to measure the side scatter and fluorescence of EVs following the manufacturer’s instructions. The instrument was calibrated for concentration and size using 200 nm PE- and AF488 fluorophore-conjugated PS beads and a Silica Nanosphere Cocktail (NanoFCM, Nottingham, England). EVs were purified using SEC and UC, as described previously. For the nano-flow cytometry experiments, EVs were resuspended in 100 µL 1X PBS. A total of 20 µL of each sample was stained with PE-conjugated α-CD63 (1:1, BD Biosciences, Franklin Lakes, NJ, USA), α-CD81 (1:1, BD Biosciences, Franklin Lakes, NJ, USA), α-NCAM1 (1:4, BD Biosciences, Franklin Lakes, NJ, USA), or α-FLAG (1:50, OriGene, Rockville, MD, USA). Dilutions indicate the antibody-to-sample volume ratio. Each sample was also co-stained with MemGlow488 (Denver, CO, USA) at a final concentration of 200 nM. Samples were incubated at 37 °C for 30 min while shaking at 180 rpm. Following incubation, the samples were diluted in 1X PBS and underwent an additional round of UC at 100,000× *g* for 1 h. Samples were then resuspended in 50 µL of 1X PBS. For nFC experiments involving HEK293T cells, the same experimental and control groups mentioned in the Western blotting section were utilized to calculate fold changes between the GDE2 and H243A groups and NT group. Fold changes were calculated between groups using the percentage of particles identified as positive for both the EV markers and MemGlow488. For experiments involving *Gde2*KO and WT neurons, the percentage of particles identified as positive for the EV markers and MemGlow488 was utilized to calculate fold changes between EVs derived from *Gde2*KO neurons as compared to WT controls. For the experiment validating the incorporation of GDE2-eFLAG into EVs, the percentage of particles identified as positive for both FLAG and the MemFlow488 was utilized to calculate the fold change between GDE2-eFLAG EVs and the non-specific binding exhibited by incubating the PE-conjugated FLAG antibody with EVs derived from WT control neurons; 3.75 × 10^6^ cells were utilized for all groups with no observable differences in cell viability or morphology. Given this, as well as the post-mitotic nature of neurons, no further normalization was performed. Three biological replicates were utilized for both groups.

### 2.11. Immunoprecipitation of GDE2-eFLAG EVs

For the immunoprecipitation assays, EVs from WT or GDE2-eFLAG+/− neuron cultures were purified from clarified conditioned culture medium using UC as described previously. Following purification, the EVs were resuspended in 1000 µL of 1X PBS. Purified EVs were then incubated with a biotinylated α-FLAG (Cell Signaling Technology, #2908, Danvers, Massachusetts, USA) at a 1:50 dilution overnight at 4 °C with gentle rotation. The following day, PierceTM Streptavidin Magnetic Beads (Thermo Fisher, 88817, Waltham, MA, USA) were prepared per the manufacturer’s instructions. Briefly, 50 µL of beads was added to one 1.5 mL tube per EV sample and inserted into a magnetic stand. A bind/wash buffer comprised of 1X tris-buffered saline with 0.1% Tween-20 Detergent was used to wash the beads prior to adding the EV–antibody mixture. The samples were incubated for 2 h at RT with gentle rocking. They were then washed with binding/wash buffer prior to lysis in GLB at 95 °C for 10 min. EV lysates were stored at −80 °C until the downstream experiments; 3.75 × 10^6^ cells were utilized for both the GDE2-eFLAG+/− neuron cultures, with no observable differences in cell viability or morphology. EVs derived from WT neurons not expressing the eFLAG tag were incubated with the biotinylated α-FLAG antibody as a control for non-specific binding.

### 2.12. Mass Spectrometry of EVs

GDE2 and H243A-transfected HEK293T cells, and WT and *Gde2*KO neurons, were cultured, and EVs were isolated by SEC and UC as described above. WT and GDE2-eFLAG^+/−^ neurons were cultured, and EVs were isolated by IP and UC as described above; 3.5 × 10^6^ HEK293T cells were utilized for a single biological replicate, and for all neuronal experiments, 3.75 × 10^6^ cells were utilized with three biological replicates. There were no observed differences in cell viability or morphology in any of the experimental conditions. The experimental and control groups utilized in these experiments are the same as the ones detailed in the previous sections, to enable the calculation of fold changes in the individual proteins identified between groups. More information on the normalization approach and fold change calculations can be found in the statistics section. All EV pellets were resuspended in GLB and run at 140 V into 10% Criterion™ TGX™ Precast Midi Protein Gels (12 + 2 well, 45 µL) (Biorad, 5671033, Hercules, CA, USA). The gels were then stained using the GelCode Blue Safe Protein Stain kit (Thermo Fisher Scientific, 1860957 (24594), Waltham, MA, USA), and labeled protein was cut from the gel. Samples were analyzed by the Taplin Biological Mass Spectrometry Facility at Harvard University. Samples were analyzed by either an Orbitrap Exploris480 mass spectrometer (Thermo Fisher Scientific, Waltham, MA, USA) or a Velos Orbitrap Elite ion-trap mass spectrometer (Thermo Fisher Scientific, Waltham, MA, USA). For both methods, excised gel bands were cut into approximately 1 mm^3^ pieces. Gel pieces were then subjected to a modified in-gel trypsin digestion procedure [37]. Gel pieces were washed and dehydrated with acetonitrile for 10 min, followed by removal of acetonitrile. The pieces were then completely dried in a speed-vac. Rehydration of the gel pieces was with 50 mM ammonium bicarbonate solution containing 12.5 ng/µL modified sequencing-grade trypsin (Promega, Madison, WI, USA) at 4 °C. After 45 min, the excess trypsin solution was removed and replaced with 50 mM ammonium bicarbonate solution to just cover the gel pieces. Samples were then placed in a 37 °C room overnight. Peptides were later extracted by removing the ammonium bicarbonate solution, followed by one wash with a solution containing 50% acetonitrile and 1% formic acid. The extracts were then dried in a speed-vac (~1 h). The samples were then stored at 4 °C until analysis.

For samples analyzed using the Orbitrap Exploris480, on the day of analysis, the samples were reconstituted in 5–10 µL of HPLC solvent A (2.5% acetonitrile, 0.1% formic acid). A nano-scale reverse-phase HPLC capillary column was created by packing 2.6 µm C18 spherical silica beads into a fused silica capillary (100 µm inner diameter x ~30 cm length) with a flame-drawn tip [38]. After equilibrating the column, each sample was loaded via a Thermo EASY-LC (Thermo Fisher Scientific, Waltham, MA, USA). A gradient was formed, and peptides were eluted with increasing concentrations of solvent B (90% acetonitrile, 0.1% formic acid). As the peptides eluted, they were subjected to electrospray ionization and then entered into an Orbitrap Exploris480 mass spectrometer (Thermo Fisher Scientific, Waltham, MA, USA). Peptides were detected, isolated, and fragmented to produce a tandem mass spectrum of specific fragment ions for each peptide. Peptide sequences (and hence protein identity) were determined by matching protein databases with the acquired fragmentation pattern by the software program Sequest (https://proteomicsresource.washington.edu/protocols06/sequest.php; Thermo Fisher Scientific, Waltham, MA, USA) [39]. All databases include a reversed version of all the sequences, and the data were filtered to between a one and two percent peptide false discovery rate.

For samples analyzed using the Velos Orbitrap Elite, on the day of analysis the samples were reconstituted in 5–10 µL of HPLC solvent A (2.5% acetonitrile, 0.1% formic acid). A nano-scale reverse-phase HPLC capillary column was created by packing 2.6 µm C18 spherical silica beads into a fused silica capillary (100 µm inner diameter x ~30 cm length) with a flame-drawn tip [38]. After equilibrating the column, each sample was loaded via a Famos auto sampler (LC Packings, San Francisco, CA, USA) onto the column. A gradient was formed, and peptides were eluted with increasing concentrations of solvent B (97.5% acetonitrile, 0.1% formic acid). As the peptides eluted, they were subjected to electrospray ionization and then entered into a Velos Orbitrap Elite ion trap mass spectrometer. Peptides were detected, isolated, and fragmented to produce a tandem mass spectrum of specific fragment ions for each peptide. Peptide sequences (and hence protein identity) were determined by matching protein databases with the acquired fragmentation pattern by the software program Sequest [39]. All databases include a reversed version of all the sequences, and the data were filtered to between a one and two percent peptide false discovery rate. All raw data have been made publicly available by uploading them to the online repository/database MassIVE at the following url: ftp://massive.ucsd.edu/v08/MSV000095349/ (accessed on 16 July 2024).

### 2.13. Statistics

Data were processed in Excel, and all statistical analysis and graphing were done on GraphPad Prism 9.3. Two-tailed independent *t*-tests or a one-way ANOVA with Tukey post hoc test for multiple comparisons were used to analyze the data, as indicated in the figure legends. For MS experiments, the data were log-transformed, normalized based on the average total signal in each group, and then the fold changes were calculated for the mean signal of each protein identified between HEK-GDE2 EVs and NT-EVs, *Gde2*KO-EV and WT-EVs, and GDE2-eFLAG EVs and control EVs captured via IP [40].

## 3. Results

### 3.1. GDE2 Overexpression Promotes EV Release in HEK293T Cells

To examine if GDE2 is capable of releasing EVs, we transfected GDE2 into Human Embryonic Kidney (HEK) 293T cells and isolated EVs from the cell culture medium 24 h later, utilizing size exclusion chromatography (SEC) followed by ultracentrifugation (UC) (Figure 1A; HEK-GDE2-EV). EVs purified from non-transfected (NT) HEK293T cells were used as a negative control (NT-EV). Transmission electron microscopy of the EV fractions confirmed the purity and integrity of the EVs isolated, thus validating this method for EV isolation (Figure 1A). The EV fractions were then analyzed by Western blot for a panel of established EV markers, including Alix [11,14] and Annexin A1 [17,41], as well as the membrane protein neural cell adhesion molecule-1 (NCAM-1) [42] (Figure 1B,C). HEK-GDE2-EVs showed a marked increase in NCAM-1 and Annexin A1 compared with NT-EVs (NCAM-1, *p* = 0.0007; Annexin A1, *p* = 0.0070) (Figure 1B,C). However, there were no significant changes in Alix expression between the HEK-GDE2-EV and the NT-EV groups (*p* = 0.4689) (Figure 1B,C). These observations suggest that GDE2 is capable of releasing EVs enriched in membrane proteins, such as Annexin A1 and NCAM-1, as opposed to general EV markers, like Alix. Additionally, Annexin A1 has been proposed as a putative marker for ectosomal biogenesis [41]. These findings are consistent with the known localization of GDE2 on the cell surface.

GDE2 is a membrane-bound enzyme that acts at the plasma membrane to cleave the GPI-anchor that tethers some proteins to the membrane [22,25,43]. To determine if the GDE2 GPI-anchor cleavage function is required for EV release, we repeated these experiments using a previously validated version of GDE2 that contains a histidine to alanine substitution at residue 243 (H243A). This point mutation abolishes the GPI-anchor cleaving function of the resultant protein, while retaining its expression at the cell surface (Appendix A). Further, H243A is expressed at similar levels to GDE2 when transfected into HEK293T cells (Appendix A). The Western blot analysis of EVs purified from HEK293T cells transfected with GDE2.H243A (H243A-EV) showed no significant changes in NCAM-1, Annexin A1, and Alix compared with NT-EVs (ns *p* > 0.05, Figure 1B,C). Thus, the ability of GDE2 to release EVs appears to require its GPI-anchor cleavage function.

### 3.2. GDE2 Overexpression Increases the Numbers and Protein Loading of Small EVs

To elucidate whether the changes in the protein markers present in the HEK-GDE2-EVs are a result of increased protein loading into the EVs or an increase in total EVs, we employed nano-flow cytometry (nFC) as an orthogonal method to measure the EV concentration and the expression of specific protein antigens simultaneously. We first analyzed the fold change in the number and size of the EVs released in NT, GDE2, and the H243A groups by measuring the total numbers and size of the EVs, assayed by staining with the lipophilic membrane probe, MemGlow 488 (Figure 2A–C). The HEK293T cells transfected with GDE2 released a significantly greater number of EVs relative to both the H243A (3.10× FC [fold change], *p* = 0.0278, Figure 2A,B) and NT (19.1× FC, *p* = 0.0017, Figure 2A,B) groups, confirming our earlier observations that GDE2 utilizes its GPI-anchor cleavage activity to release EVs (Figure 1B,C). Additionally, we found that EVs released by the GDE2 group showed a small but significant increase in size relative to the NT group (78 nm and 69 nm, respectively; *p* = 0.0098) but not the H243A group (Figure 2A,C). The overwhelming majority of the EVs released by GDE2 exhibited a diameter less than 200 nm, as measured by TEM and nFC (Figure 1A and Figure 2A), thus this population of EVs is classified as small EVs (sEVs) in accordance with MISEV guidelines [13].

We next analyzed the percentage of total EVs (MemGlow488+) that express known EV antigens, to determine if GDE2 is required for protein loading into the EVs. We examined NCAM-1, as this protein showed the greatest increase in HEK-GDE2-EV fractions (Figure 1B,C) and the tetraspanins CD63 and CD81, which are membrane-bound markers of EVs [10,14]. The percentages of NCAM-1+, CD63+, and CD81+ EVs were significantly increased in HEK-GDE2-EVs relative to NT-EV (NCAM-1:1.56× FC, *p* < 0.0001; CD63: 1.90× FC, *p* = 0.0007; CD81: 1.99× FC, *p* = 0.0486), suggesting that GDE2 increases the loading of all three proteins into the EVs (Figure 2D–I). To examine if the loading of NCAM-1, CD63, and CD81 by GDE2 depends on its GPI-anchor cleavage activity, we repeated the experiment using the GDE2 catalytic mutant H243A. Interestingly, the percentage of NCAM1+ and CD63+ EVs was also significantly increased in H243A-EVs relative to NT-EV (NCAM-1: 1.37× FC, *p* = 0.0006; CD63: 1.59× FC, *p* = 0.0105), while the percentage of CD81+ EVs showed a trending but not significant increase between the groups (1.71× FC, *p* = 0.1661) (Figure 2D–I). These observations suggest that the increase in these markers present in the EVs is due to an alternative function of the GDE2 protein that is separate from its GPI-anchor cleavage activity. However, we detected a moderate but significant decrease in the percentage of NCAM1+ EVs between the HEK-GDE-EV and H243A-EV groups (1.14× FC, *p* = 0.0418), suggesting that GDE2 GPI-anchor cleavage may also contribute to the loading of select cargoes into EVs (Figure 2D,E). Taken together, these observations suggest that GDE2 expression increases the number of sEVs through GPI-anchor cleavage and affects the loading of select cargoes via various mechanisms that include non-enzymatic and enzymatic functions.

### 3.3. GDE2 Influences sEV Release in Primary Cortical Neurons

GDE2 is primarily expressed in neurons in the embryonic and adult nervous systems [22,43]. To determine if GDE2 is required for the release of and/or protein loading into EVs from neurons at physiological expression levels, we analyzed EVs purified from cortical neurons from mice lacking GDE2 (*Gde2*KO) and WT controls that had been cultured for 14–17 days in vitro (DIV).

We collected conditioned medium from DIV14–17 cultured WT and *Gde2*KO neurons, purified EVs using the same approach as described for the HEK293T experiments, i.e., sequential SEC and UC, and analyzed them by nFC (Figure 3A). MemGlow488 was similarly utilized to co-stain the EVs. An analysis of total EV numbers showed that the number of EVs released by the *Gde2*KO neurons was significantly lower than in WT neurons (−0.4× FC, *p* = 0.0123), indicating that GDE2 contributes to the release of EVs from neurons (Figure 3B,C). Unlike the increase in EV size observed between the HEK-GDE2-EVs and NT-EVs (Figure 2A,C), there was no significant difference in the size of the *Gde2*KO EVs (mean size = 76.8 nm) relative to the WT EVs (mean size = 82.5 nm) (Figure 3B,D). However, given that the EV size was less than 200 nm, these EV populations would also be classified as sEVs. Combined with our observations in HEK293T cells, these findings suggest that GDE2 is necessary and sufficient for the release of sEVs from neurons.

We next utilized the same nFCM panel of EV markers (CD63, CD81, NCAM-1) to investigate the influence of GDE2 on protein loading in neuronal EVs (Figure 3E–J). Surprisingly, none of the EV markers analyzed in our panel were significantly changed between the *Gde2*KO-EV and WT-EV groups (CD63: *p* > 0.9999, CD81: *p* = 0.6679, NCAM-1: *p* = 0.1469) (Figure 3E–J). Given that these proteins showed changes in response to GDE2 overexpression in HEK293T cells, it is possible that the extent to which these specific antigens are loaded into EVs may be stoichiometrically dependent on the amount of GDE2 expressed. Alternatively, other compensatory mechanisms may operate in neurons to ensure the appropriate loading of these cargoes in EVs in the absence of GDE2.

### 3.4. GDE2 Releases Neuronal sEVs Enriched in Putative Ectosomal Markers

Our experiments investigating the influence of GDE2 on EV release using overexpression and *Gde2*KO neuronal models indicate a clear influence of GDE2 on sEV release (Figure 1, Figure 2 and Figure 3). A deeper understanding of the protein content of sEVs released by GDE2 will help determine protein markers that identify a molecular signature of the EV subtypes released by GDE2 and, additionally, may provide insight into the putative functions of these EVs. Accordingly, we utilized mass spectrometry to elucidate the total protein cargoes of HEK-GDE2-EVs. GDE2.H243A-EVs and NT-EVs served as negative controls. In parallel, we surveyed the protein cargoes in EVs released in WT and *Gde2*KO cultured neurons. In the HEK293T overexpression model, 318 proteins showed a greater expression or were uniquely found in the HEK-GDE2-EVs relative to NT-EVs (Figure 4A,B, Appendix A). In the neuronal model, 202 proteins showed a reduced expression or were absent in the *Gde2*KO-EVs relative to the WT-EVs, and 172 proteins showed a greater expression or were only found in the *Gde2*KO-EVs compared with the WT-EVs (Figure 4A,C, Appendix A). Moreover, 41 proteins were found to have an increased expression in the HEK-GDE2-EVs relative to the NT-EVs and decreased expression in the *Gde2*KO-EVs relative to WT-EVs (Figure 4A–C, Appendix A). Interestingly, GDE2 was the top protein detected in HEK-GDE2-EVs purified from HEK293T cells relative to NT-EVs with a 9.07× FC but was markedly reduced in GDE2.H243A-EVs. Although GDE2 release was not detected in the neuronal model in this paradigm, later experiments confirm its release in neuronal EVs (see Figure 6B). This observation suggests that GDE2 itself is released in EVs and that this depends on its GPI-anchor cleavage function (see Figure 6B and Appendix A).

A Gene Ontology (GO) term analysis using the DAVID database was utilized to identify the cellular components of the proteins showing greater expression in HEK-GDE2-EVs relative to NT-EVs and decreased or absent in the *Gde2*KO-EV group. The terms extracellular vesicle, extracellular exosome, and extracellular organelle were found for both groups (Figure 4D, Appendix A). Among the top changed proteins in this category for both groups were the established EV markers, TSG101 and SDCBP, which were both increased in the HEK-GDE2-EVs and decreased in the *Gde2*KO-EVs (Figure 4E, Appendix A). Notably, CD9 was identified as being exclusively decreased in the *Gde2*KO-EVs, which was the greatest change out of all the identified proteins (Figure 4E, Appendix A). Given our nFC data indicating a small EV size (Figure 3D) and recent work by Mathieu et al., 2021 [16] suggesting that CD9 may be predominantly released in small ectosomes, it is possible that GDE2 expression influences this population of EVs. Work by this same group also identified the membrane proteins BSG, SLC3A2, and PTGFRN as potential small ectosomal markers. BSG was decreased in the *Gde2*KO-EVs, whereas SLC3A2 and PTGFRN showed little change (Figure 4E, Appendix A). However, SLC1A6, SLC38A3, SLC39A8, SLC44A2, and SLC6A11 were all absent in the *Gde2*KO-EVs, and SLC6A1, SLC1A5, and SLC16A1 were both decreased in the *Gde2*KO-EVs (Figure 4E, Appendix A). Taken together, these observations suggest that GDE2 is required for the release of neuronal sEVs that contain TSG101 and SDCBP, as well as the putative ectosomal markers CD9 and BSG.

### 3.5. Neuronal sEVs Released by GDE2 Imply Possible Roles in Synaptic and Redox Function

Next, we expanded our search to identify novel protein signatures in the sEVs purified from GDE2-overexpressing HEK293T cells and WT and *Gde2*KO cultured neurons to gain insight into their potential ontogeny and function. A GO term analysis using the DAVID database was utilized to identify the cellular components of the proteins showing greater expression in the HEK-GDE2-EVs relative to NT-EVs and decreased or absent in the *Gde2*KO-EV group. Interestingly, the top cellular component GO term found for both groups was synapse, with others, such as neuron part, post-synapse, and glutamatergic synapse, also appearing in the top terms (Figure 5A,B, Appendix A). Given that synapse was the top cellular component for both the HEK-GDE2-EV upregulated proteins and the *Gde2*KO-EV downregulated proteins, we investigated the protein sets identified in this category further. The solute carrier proteins SLC6A1 and SLC16A1 were both decreased in the *Gde2*KO-EVs, whereas SLC1A6 and SLC6A11 were absent altogether (Figure 5C, Appendix A). These proteins are collectively responsible for transporting both excitatory (glutamate, SLC1A6) and inhibitory (GABA, SLC6A1, and SLC6A11) neurotransmitters, as well as metabolites (SLC16A1), across the synaptic membrane to modulate synaptic activity [44,45,46,47]. Proteins known to be involved in Alzheimer’s disease progression, (such as CLU, APOE, and ADAM10) and Parkinson’s Disease (such as PARK7 and VPS35) were also identified [48,49,50,51], suggesting a potential role of GDE2-released EVs in neurodegeneration (Figure 5C, Appendix A).

We further analyzed the sets of identified proteins by annotating them with GO terms for molecular and biological function. Under the molecular function annotation, cytoskeletal protein binding was a top term identified for both groups (Figure 5D,E; Appendix A). Proteins identified in these categories include ACTR2 and 3, ANXA6, GSN, MSN, and WDR1 (Appendix A). Under the biological process annotation, the GO terms identified were predominantly related to the regulation and organization of the cytoskeleton, including the terms cytoskeleton organization and actin cytoskeleton organization (Appendix A). Proteins identified in these categories for both groups include ITGB1, GSN, DYNC1H1, and WDR1, with APOA1 decreased in the *Gde2*KO-EVs but not increased in the HEK-GDE2-EVs (Appendix A). Additionally, ARF1 and ICAM1 were absent from both the HEK-GDE2-EVs and *Gde2*KO-EVs (Appendix A). Previous studies suggest that ectosomal populations can be generated through manipulation of the plasma membrane via modulation of the actin cytoskeleton [14,17,28]. Accordingly, these observations suggest that GDE2 might regulate the production of small ectosomes by regulating cytoskeletal dynamics.

Interestingly, GO terms associated with cellular responses to oxidative stress, such as antioxidant activity and peroxidase activity, were identified for the proteins that were decreased in the *Gde2*KO-EVs but were not found for those increased in the HEK-GDE2-EVs (Figure 5D,E; Appendix A). The set of proteins identified in this category included the peroxiredoxins PRDX1, PRDX2, PRDX4, PRDX5, and PRDX6, as well as APOE, CP, PARK7, and SOD1 (Figure 5F, Appendix A). These results suggest that GDE2-released sEVs may play a role in modulating intercellular responses to oxidative stress, which is often dysregulated in various neurodegenerative diseases [52]; however, this requires further validation.

### 3.6. Proteomic Signatures of sEV Subtypes Containing GDE2

Our proteomic analysis in HEK293T cells reveals that GDE2 can be released in sEVs (Figure 4, Appendix A), suggesting that GDE2 may release different populations from neurons, some of which may include GDE2 as cargo. Although we did not detect GDE2 in sEVs released from neurons, it is possible that this population is less abundant and was not well represented in our initial purification. To address these possibilities, we first developed a strategy to label GDE2 by introducing a 3XFLAG tag in the third extracellular loop of GDE2 between transmembrane domains 5 and 6 (GDE2-eFLAG). The insertion of the tag at this extracellular location would enable the immunoprecipitation of EVs expressing the hybrid GDE2-FLAG protein for targeted proteomic analysis (Figure 6A). The introduction of the 3XFLAG did not impair GDE2 GPI-anchor cleavage activity or release function (Figure 6B and Appendix A). Additionally, the transfection of GDE2-eFLAG into HEK293T cells, followed by staining with anti-FLAG antibody and co-staining with MemGlow488 without prior permeabilization, confirmed that GDE2-eFLAG was correctly transported to the cell membrane and is located at the cell surface (Appendix A). Using the same tagging strategy, we generated a novel mouse line using CRISPR/Cas9 to genetically label endogenous GDE2 (see Materials and Methods).

To determine if endogenous GDE2 is released in sEVs, we generated primary cortical neuronal cultures from P0-1 WT or GDE2-eFLAG^+/−^ mice, cultured them for 14–17 DIVs, and purified EVs from the conditioned medium using SEC and UC. WT-EVs and GDE2-eFLAG EVs were then stained with a PE-conjugated FLAG antibody and MemGlow488 and subjected to nFC (Figure 6B). Using this method, we were able to detect a significantly greater signal in the PE channel for the GDE2-eFLAG EVs than from WT-EVs, confirming that endogenous GDE2-eFLAG is indeed released in EVs from neurons (Figure 6B) (5.71x FC, *p* = 0.0335). After correction for the signal attributed to the nonspecific binding of the FLAG antibody, these results indicated 29.1% of EVs released by GDE2-eFLAG^+/−^ cultured neurons contained GDE2. To examine if EVs containing GDE2 itself differ from other EVs, we purified EVs from the conditioned medium of WT or GDE2-eFLAG^+/−^ DIV14-17 cortical neuronal cultures as described above and then utilized a biotinylated FLAG antibody and streptavidin-coated magnetic beads to immunoprecipitate EVs for targeted proteomic analysis. WT-EVs served as a control for non-specific binding, as this group still showed substantial signals in the nFC analysis (Figure 6B). Overall, 177 proteins were detected in the GDE2-eFLAG group after comparison with the WT group (Figure 6C, Appendix A). To determine if the cargo of GDE2-eFLAG EVs differed from the cargo of EVs released by GDE2 in neurons, we compared the GDE2-eFLAG EV proteome with the proteins decreased in the *Gde2*KO-EVs. A total of 57 proteins were shared between GDE2-eFLAG EVs and EVs released by GDE2, with 106 proteins uniquely detected in the GDE2-eFLAG cohort and 117 proteins uniquely expressed in EVs released by GDE2 (Figure 6C, Appendix A). These observations support the idea that GDE2 is capable of releasing different populations of sEVs with different cargoes, one population of which includes EVs containing GDE2 itself.

To gain deeper insight into the potential roles of sEVs containing GDE2, we utilized the database DAVID to annotate the proteins with GO terms for cellular components and molecular functions. Similar to our datasets from the proteins found to be decreased in the *Gde2*KO-EVs, the top three cellular components for the proteins identified in the GDE2-eFLAG EVs were the terms extracellular region and extracellular region part and extracellular space (Figure 6D, Appendix A). Additionally, a cluster of neuron-specific cellular components was also identified among the top terms for the GDE2-eFLAG EVs, including synapse, neuron part, synapse part, neuron projection, and presynapse (Figure 6D, Appendix A). Consistent with our earlier results, an analysis of annotations for molecular function found the term cytoskeletal protein binding, further supporting the role of cytoskeletal protein organization in the mechanism of biogenesis for GDE2-released sEVs (Appendix A). Interestingly, another cluster of terms similar to our earlier results comprised terms related to the cellular response to oxidative stress, including antioxidant activity, peroxidase activity, oxidoreductase activity, oxygen binding, and peroxiredoxin activity (Figure 6E, Appendix A). Of note, all these terms were also identified in the top cellular components and molecular functions for the set of proteins that was decreased in the *Gde2*KO-EVs (Figure 4 and Figure 5), indicating that these terms were conserved across the sets of proteins decreased in the *Gde2*KO-EVs and those present in the GDE2-eFLAG EVs.

### 3.7. Comparative Analysis of Protein Cargoes in sEVs Released by GDE2 and GDE2-EVs

Given that the GO terms pertaining to EVs, synapses, and redox function were conserved in neuronal sEVs released by GDE2 and in GDE2-eFLAG-EVs, we investigated the proteins in these categories to elucidate the protein signatures exclusively found in GDE2-containing EVs (GDE2-eFLAG+), exclusively decreased in the *Gde2*KO-EVs, or identified in both EV populations. In the extracellular category, we identified CD9 and BSG as being exclusively decreased in the *Gde2*KO-EVs, suggesting that it is present in the GDE2-released EV population that does not contain GDE2 itself (Figure 6F, Appendix A). Further, we identified a set of extracellular annotated proteins present in both populations that included the general EV proteins SDCBP and TSG101 (Figure 6G, Appendix A). Thus, GDE2 releases sEVs that fall into two molecularly distinct groups: Group I (SDCBP, TSG101, CD9, BSG)+ and Group 2 (SDCBP, TSG101, GDE2)+.

Through further investigation of the synaptic annotated protein sets, we could identify proteins involved in modulating synaptic activity that were exclusively expressed in each EV population and those present in both EV populations (Figure 6H,I, Appendix A). In the set of proteins exclusively decreased in the *Gde2*KO-EVs, we identified proteins involved in the modulation of synaptic activity, such as solute carrier proteins SLC6A1, SLC16A1, and SLC6A11 (Figure 6H, Appendix A) [44,45,46,47]. Proteins exclusively in the GDE2-eFLAG EVs included ATP1A1 and ATP1A3, which modulate synaptic ion transport (Figure 6G, Appendix A) [53]. ATP1A3 is of note as it was recently shown to be enriched in neuronal EVs [54]. Another protein exclusively identified in the GDE2-eFLAG EVs is the presynaptic protein SYP, which regulates synaptic vesicle release (Figure 6H, Appendix A) [55]. Proteins identified under synaptic protein annotation and conserved across the two EV populations included SDCBP, APOE, CLU, ITGB1, and RAB10 (Figure 6I, Appendix A).

An investigation of proteins under the annotation terms associated with the cellular response to oxidative stress showed that PRDX1, PRDX5, PARK7, and SOD1 were exclusively decreased or absent in the *Gde2*KO-EVs and were not present in the GDE2-eFLAG EVs (Figure 6J, Appendix A). The antioxidant protein CAT was the only one identified in this category as being exclusively in GDE2-eFLAG EVs but not decreased in the *Gde2*KO EVs (Figure 6J, Appendix A). Interestingly, the peroxiredoxins PRDX2, PRDX4, and PRDX6, in addition to APOE and CP, were found in both EV populations (Figure 6K, Appendix A). These results suggest that the two GDE2-released sEV populations carry functional cargoes involved in redox biology that contain different and conserved proteins between the two populations.

## 4. Discussion

GDE2 is one of three six-transmembrane proteins with an extracellular enzymatic domain that acts at the cell surface to cleave the GPI-anchor that tethers some proteins to the membrane [22,43]. We show here that GDE2 is sufficient for releasing sEVs from the plasma membrane and that this release activity depends on its GPI-anchor cleavage function. Furthermore, GDE2 is also required for the loading of select cargoes into sEVs, and this process utilizes enzymatic and non-enzymatic mechanisms. In neurons, GDE2 is required for the release of different sEV populations that can be stratified into two broad classes according to the presence of GDE2 itself. Proteomic profiling, followed by GO term analyses, reveals shared terms across both classes of EVs that correspond to cytoskeletal organization, synaptic compartments, redox signaling, and oxidoreductase and antioxidant activity, suggesting a potential mechanism for GDE2-dependent EV biogenesis and roles for GDE2 EVs in synaptic and redox biology. These observations identify GDE2 as a novel regulator of sEV populations from neurons and raise the possibility that the sEVs released by GDE2 may be required for synaptic and redox-dependent mechanisms important for neuronal function and survival. However, these potential functions for sEVs released by GDE2 require further validation and functional evaluation.

Our studies identify GDE2 as a physiological pathway involved in releasing at least two molecularly distinct populations of sEVs from neurons. EVs released by GDE2 are not enriched with the general EV marker Alix [11,13,14] but instead contain the putative ectosomal marker Annexin A1 [13,17,41]. A proteomic profiling of GDE2-dependent EVs shows that they share the general EV markers TSG101 [11,13,14] and SDCBP [55,56] and also identifies protein signatures consistent with at least two populations of sEVs: one that contains GDE2 and a second that lacks GDE2 but contains CD9 and BSG. GDE2 activity relies on its ability to cleave the GPI-anchor, a posttranslational modification that tethers some proteins to the membrane [22,25,43]. We show here that the GDE2 EV release function requires its GPI-anchor cleavage activity, which is consistent with its involvement in the release of ectosomes, given that GDE2’s ability to cleave GPI-anchors occurs solely on the cell surface [57]. Our studies also show that GDE2 influences protein loading into EVs and involves mechanisms dependent and independent of its enzymatic function. Of note, we identified the GPI-anchored proteins LPL [58] and NTM [59] in GDE2-released EVs affected by *Gde2*KO, raising the possibility that GDE2 may also release GPI-anchored proteins via this mechanism. Future investigations of the mechanisms involved will deepen our understanding of how different cargoes are selectively incorporated into EVs and may identify novel non-enzymatic functions for GDE2.

Our discovery that GDE2’s ability to release sEVs requires its enzymatic activity suggests that it does so via the cleavage and regulation of a GPI-anchored substrate expressed on the cell surface of neurons. The identity of this GPI-anchored substrate and the mechanism by which GDE2 promotes sEV release remains to be determined. Previous studies have suggested that the remodeling of the actin cytoskeleton is involved in the generation of ectosomal populations (often referred to as microvesicles) [2,14,28,41]. Interestingly, one of the top GO terms for molecular and biological function identified in our proteomic profiling of ectosomes released by GDE2 involved cytoskeletal proteins and actin modulation. Moreover, GDE3, which is closely related to GDE2 [20,22,43], was recently shown to drive the production of a specific ectosome (microvesicle) population from astrocytes through the regulation of the actin cytoskeleton via its interaction with WAVE3, a component of the WAVE Regulatory Complex (WRC), a major regulator of actin remodeling [28]. These observations raise the possibility that the six-transmembrane GDE proteins might regulate the production of EVs of ectosomal origin via remodeling of the cytoskeleton. One notable difference between the GDE2 and GDE3 EV release function is that while GDE2 EV release activity is dependent upon its enzymatic activity, GDE3 EV release activity requires its intracellular N-terminal domain but not its catalytic activity [20,28]. Thus, while GDE2 and GDE3 may converge on the cytoskeleton to regulate EV release, their mechanisms of action are likely to be different.

Our analyses of the cargoes of the sEV populations released by GDE2 identified a top annotation cluster consisting of GO terms related to the synaptic cellular compartment. We identified several proteins decreased in the *Gde2*KO-EVs involved in the modulation of synaptic function, such as a number of solute carrier proteins involved in the transport of excitatory and inhibitory neurotransmitters and metabolites across the synaptic membrane to modulate synaptic activity, as well as those involved in the regulation of ion transport, such as CAMK2D [60] and various ATPase subunits [61]. Further, ATP1A3, which was recently shown to be altered in EVs purified from AD patient biofluid samples [54], was exclusively found in the GDE2-eFLAG EVs, along with the presynaptic protein, SYP, which regulates synaptic vesicle release [62]. Mice lacking GDE2 exhibit synaptic abnormalities and display cognitive deficits in behavioral tests, warranting further testing to determine if EVs released by GDE2 regulate aspects of synaptic function [32]. Interestingly, EVs released by GDE3 have been shown to modulate post-synaptic activity, which may suggest conserved roles for EVs released by the six-transmembrane GDEs in synapse biology [28]. We note that both GDE2 and GDE3 themselves are released in EVs [28], but the significance of this remains unclear. GDE3 retains its ability to cleave and release GPI-anchored proteins from the EV membrane. Accordingly, we speculate that the presence of six-transmembrane GDEs in EVs may constitute a stable signaling hub that could play roles in intercellular communication. Alternatively, GDE2 and GDE3 may serve as molecular scaffolds to recruit select proteins into EVs. Support for this scaffolding function comes from our observations that GDE2-eFLAG EVs contain cargoes that are distinct from GDE2-released EVs and that some of these cargoes include proteins such as Prdx1 and Prdx4 that are known to interact with GDE2 [63,64]. However, both these possibilities remain to be tested.

In addition to our identification of synaptic terms in our GO term analysis of GDE2-released sEVs, we identified a cluster of terms annotated under molecular function pertaining to cellular redox. Within this cluster, we identified SOD1 and the peroxiredoxin proteins PRDX1, PRDX2, PRDX4, PRDX5, and PRDX6, as either decreased or absent in the *Gde2*KO-EVs relative to WT-EVs, whereas PRDX2, PRDX4, and PRDX6 and the antioxidant protein CAT1 were found in GDE2-eFLAG-EVs. These proteins are critical for neuronal redox homeostasis and have been shown to be dysregulated in neurodegenerative diseases, such as AD and ALS [65,66,67]. *Gde2*KO mice show an age-progressive neurodegeneration and gliosis indicative of increased inflammation, and GDE2 distribution and function are shown to be disrupted in AD, ALS, and ALS/FTD [23,26,27]. We speculate that sEVs released by GDE2 may have functions in the regulation of the redox state in neurons and that the failure of GDE2 EV release may contribute to neurodegeneration. Whether this is the case requires further investigation. We note that GDE2 trafficking to the plasma membrane and its activity at the cell surface of neurons during embryonic development are regulated by PRDX4 and PRDX1, respectively, which may also suggest roles for GDE2-EVs in redox-dependent mechanisms in neuronal development [63,64]. Potential interactions between GDE2 and PRDX proteins, and how their release in EVs might impact neuronal development and survival through redox-dependent mechanisms, warrant further investigation.

## 5. Conclusions

Altogether, our studies indicate that GDE2 is capable of releasing numerous sEV populations with different protein cargoes that are implicated in modulating synaptic activity and cellular responses to oxidative stress, many of which are involved in neurodegenerative diseases. Previous studies have shown that GDE2 aberrantly accumulates in intracellular compartments in the postmortem brain of patients with AD, ALS, and ALS/FTD [31]. A biochemical fractionation of postmortem brain from patients with AD and the proteomic analysis of the CSF of patients with ALS show a reduction in the release of GPI-anchored proteins consistent with disrupted GPI-anchor cleavage activity [31]. Combined with our studies here, these observations raise the compelling notion that the GDE2 release of sEVs may be compromised in select neurodegenerative diseases and that this failure could contribute to the disease pathology. Consistent with this hypothesis, an analysis of published datasets of EV cargoes from patient populations reveals similarities to our proteomic profiles of *Gde2*KO-EVs [68]. For example, GNAO1 and SLC6A1 are reduced in EVs purified from the tissue and CSF of patients with AD and are also found to be reduced in *Gde2*KO-EVs [68]. Further, BASP1 and ATP2B2 were found to be increased in AD patient EVs and *Gde2*KO-EVs [68]. Future studies geared towards clarifying the functional roles of EVs released by GDE2 are needed to determine if they have roles in synaptic and redox biology and how they might contribute to neurodegenerative disease pathogenesis, which may have diagnostic and prognostic implications.

## Figures and Tables

**Figure 1 cells-13-01414-f001:**
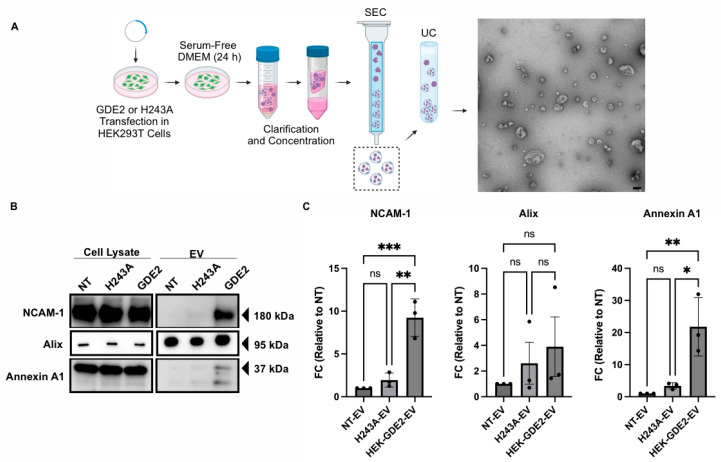
GDE2 overexpression promotes EV release in HEK293T cells. (**A**) Schematic of the workflow used to produce, purify, and analyze EVs from HEK293T cells, along with representative TEM image of purified EVs (scale bar = 100 nm). (**B**) Representative Western blots displaying markers, NCAM-1, Alix, and Annexin A1 in cell lysates and EVs from transfected HEK293T cells. (**C**) Quantification of fold change (FC) for each marker analyzed, normalized to signal derived from corresponding cell lysate. DMEM, Dulbecco’s modified Eagle medium; EV, extracellular vesicle; HEK293T, Human Embryonic Kidney 293T; NCAM-1, neural cell adhesion molecule-1; NT, non-transfected; SEC, size exclusion chromatography; UC, ultracentrifugation; ns *p* > 0.05, * *p* < 0.05, ** *p* < 0.01; *** *p* < 0.001; mean ± SEM, one-way ANOVA with Tukey’s multiple comparisons test. N = 3 biological replicates.

**Figure 2 cells-13-01414-f002:**
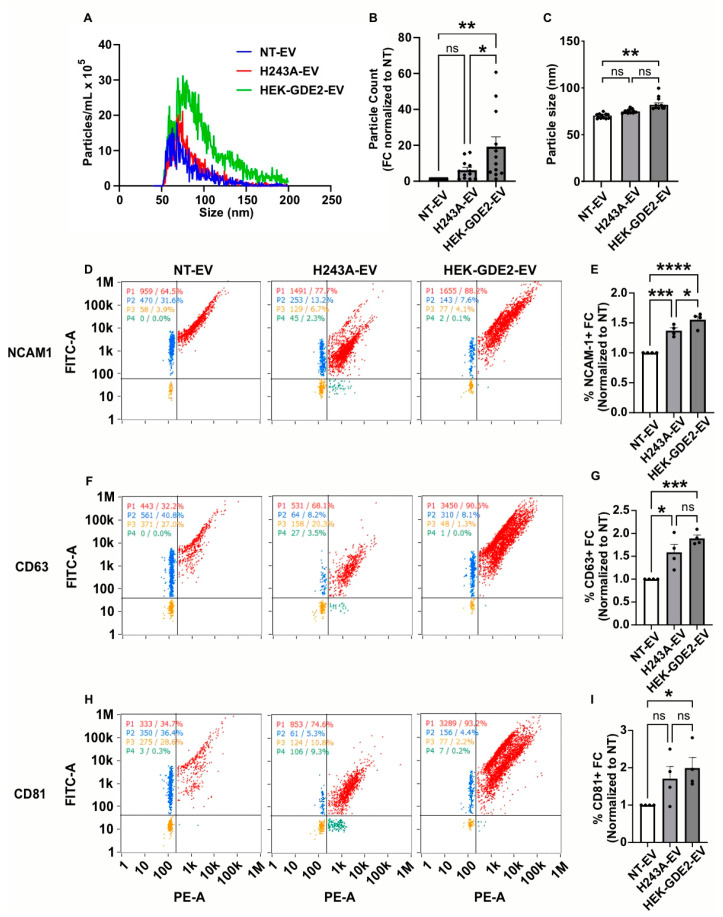
GDE2 overexpression increases sEV numbers and protein loading. (**A**) Representative histogram depicting both size and concentration of EVs measured via nFC for NT (blue), H243A (green), and GDE2 (red) groups. (**B**) Quantification of EV count fold change. (**C**) Quantification of EV size. (**D**) Representative scatter plots for NCAM-1 (blue = lipid membrane positive, antigen negative; red = lipid membrane positive, antigen positive; orange = lipid membrane negative, antigen negative; green = lipid membrane negative, antigen positive). (**E**) Quantification of percentage of EVs positive for NCAM-1 relative to total lipid membrane positive signal, indicating increased incorporation of NCAM-1 into both H243A and HEK-GDE2-EVs relative to NT-EVs, but to greater extent in HEK-GDE2-EV group. This analysis focuses primarily on the loading of NCAM-1 into EVs and accordingly does not contradict results in Figure 1 that detect an increase in EVs in response to GDE2. (**F**) Representative scatter plots for CD63. (**G**) Quantification of percentage of EVs positive for CD63. (**H**) Representative scatter plots for CD81. (**I**) Quantification of percentage of EVs positive for CD81. nFC, nano-flow cytometry; NT, non-transfected. ns *p* > 0.05, * *p* < 0.05; ** *p* < 0.01; *** *p* < 0.001; **** *p* < 0.0001; mean ± SEM, one-way ANOVA with Tukey’s multiple comparisons test. N = 3–4 biological replicates.

**Figure 3 cells-13-01414-f003:**
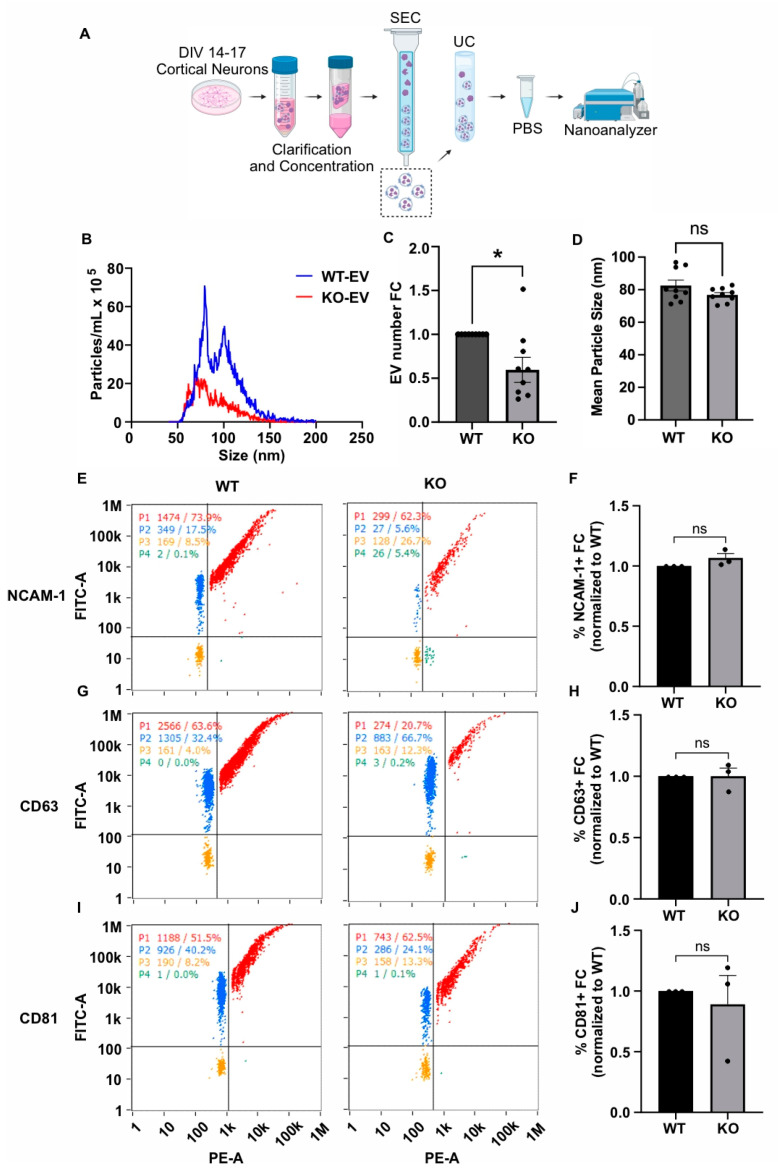
GDE2 is required for sEV release in primary cortical neurons. (**A**) Schematic of workflow to produce, purify, and analyze EVs from DIV14–17 P0-1 primary cortical neurons from either WT or *Gde2*KO mice (KO). (**B**) Representative histogram depicting size and concentration of *Gde2*KO-EVs (red) relative to WT-EVs (blue). (**C**) Quantification of EV count fold change. (**D**) Quantification of mean EV size. (**E**) Representative scatter plots for NCAM-1 (blue = lipid membrane positive, antigen negative; red = lipid membrane positive, antigen positive; orange = lipid membrane negative, antigen negative; green = lipid membrane negative, antigen positive). (**F**) Quantification of percentage of EVs positive for NCAM-1 relative to total lipid membrane positive signal. (**G**) Representative scatter plots for CD63. (**H**) Quantification of percentage of EVs positive for CD63. (**I**) Representative scatter plots for CD81. (**J**) Quantification of percentage of EVs positive for CD81. DIV, days in vitro; SEC, size exclusion chromatography; UC, ultracentrifugation. ns *p* > 0.05, * *p* < 0.05; mean ± SEM, unpaired two-tailed *t*-test, N = 3 biological replicates.

**Figure 4 cells-13-01414-f004:**
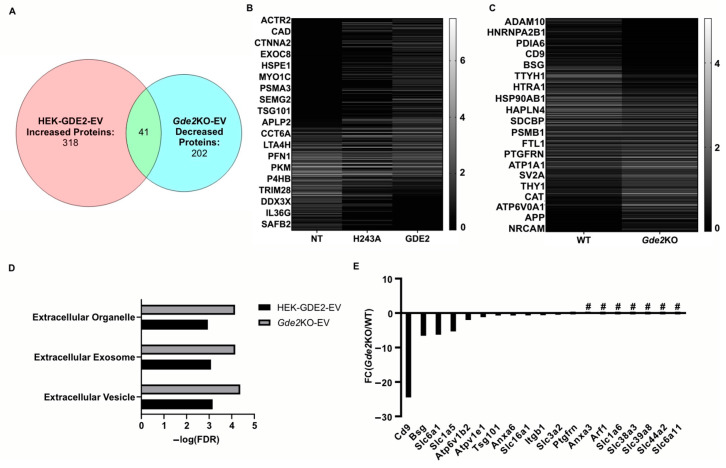
GDE2 releases EVs enriched in ectosomal markers. (**A**) Venn diagram depicting the number of proteins increased in HEK-GDE2-EVs (left), number of proteins identified as being decreased in *Gde2*KO-EVs (right), and number of similar proteins between the two groups (middle). (**B**) Representative heatmap depicting proteins identified in EVs purified from GDE2, H243A, or non-transfected HEK293T cells. (**C**) Representative heatmap depicting proteins identified in EVs purified from *Gde2*KO or WT EVs. (**D**) Extracellular annotated GO terms identified for proteins found to be increased in HEK-GDE2-EVs (black) or decreased in *Gde2*KO-EVs (gray). (**E**) Graphical representation of fold changes for proteins found to be decreased or absent in the *Gde2*KO-EVs relative to WT-EVs. EV, extracellular vesicle; FC, fold change; FDR, false detection rate; NT, non-transfected; # = absent from *Gde2*KO-EVs. N = 3 biological replicates for neuron experiments, N = 1 for HEK293T cell experiment.

**Figure 5 cells-13-01414-f005:**
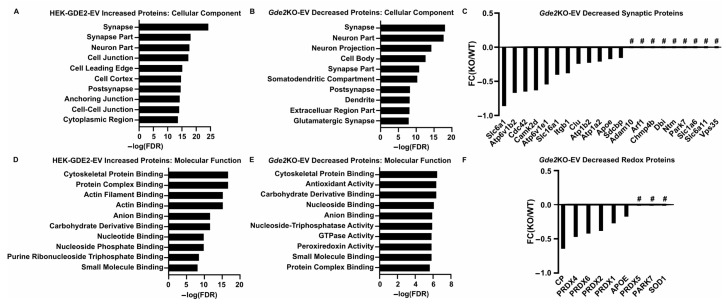
Neuron-enriched and functional cargoes of GDE2-released sEVs. (**A**) Top cellular component GO terms found for proteins identified in HEK-GDE2-s and (**B**) *Gde2*KO-EVs. (**C**) Graphical representation of fold changes for top proteins either decreased or absent in *Gde2*KO-EVs relative to WT-EVs. (**D**) Top molecular function GO terms found for proteins identified in HEK-GDE2-EVs and (**E**) *Gde2*KO-EVs. (**F**) Graphical representation of fold changes for proteins either decreased or absent in *Gde2*KO-EVs relative to WT-EVs and annotated with molecular functions related to oxidative stress responses. FC, fold change; FDR, false detection rate; # = absent in *Gde2*KO-EVs. N = 3 biological replicates for neuron experiments, N = 1 for HEK293T cell experiment.

**Figure 6 cells-13-01414-f006:**
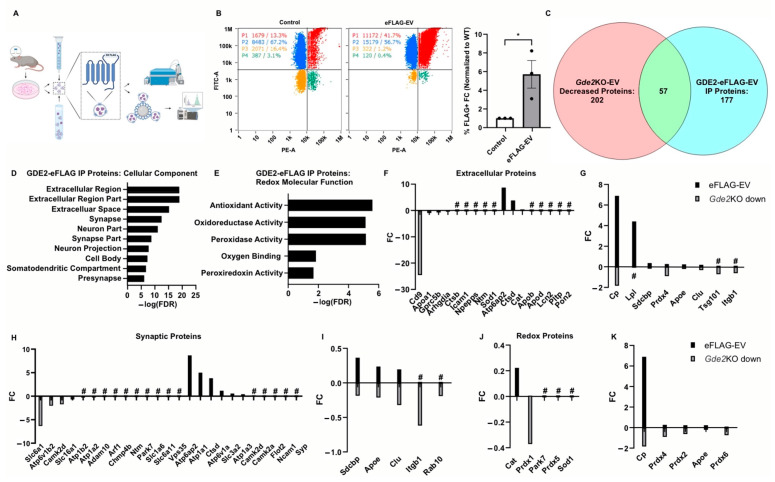
Proteomic signatures of sEV subtypes released by GDE2. (**A**) Graphical representation of the workflow for purifying EVs, immunoprecipitation, and analysis via nFC or mass spectrometry. (**B**) Representative scatter plots for WT (left) and GDE2-eFLAG (middle) stained with PE-conjugated FLAG antibody and quantification of percentage of EVs detected as FLAG+ (right). (**C**) Venn diagram depicting the number of proteins decreased in *Gde2*KO-EVs (left), detected above background in the GDE2-eFLAG EVs (right), and similar proteins between the two groups. (**D**) Top cellular component GO terms for proteins in GDE2-eFLAG EVs. (**E**) Molecular function GO terms for GDE2-eFLAG EVs related to redox biology. (**F**) Fold changes for proteins decreased in *Gde2*KO-EVs (gray) or present in GDE2-eFLAG EVs (black) annotated for extracellular component GO terms. (**G**) Fold changes for proteins both decreased in *Gde2*KO-EVs (gray) and present in GDE2-eFLAG EVs (black) annotated for extracellular component GO terms. (**H**) Fold changes for proteins decreased in *Gde2*KO-EVs (gray) or present in GDE2-eFLAG EVs (black) annotated for synaptic cellular component GO terms. (**I**) Fold changes for proteins both decreased in *Gde2*KO-EVs (gray) and present in GDE2-eFLAG EVs (black) annotated for synaptic cellular component GO terms. (**J**) Fold changes for proteins decreased in *Gde2*KO-EVs (gray) or present in GDE2-eFLAG EVs (black) annotated for molecular functions related to redox biology. (**K**) Fold changes for proteins both decreased in *Gde2*KO-EVs (gray) and present in GDE2-eFLAG EVs (black) annotated for molecular functions related to redox biology. EV, extracellular vesicle; FC, fold change; FDR, false detection rate; nFC, nano-flow cytometry. * *p* < 0.05; # = absent from *Gde2*KO-EVs or GDE2-eFLAG background. Mean ± SEM, unpaired two-tailed *t*-test, N = 3 biological replicates.

## Data Availability

The original contributions presented in the study are included in the article/Appendix A; further inquiries can be directed to the corresponding author. All raw proteomic data have been made publicly available by uploading them to the online repository/database MassIVE at the following url: ftp://massive.ucsd.edu/v08/MSV000095349/ (accessed on 16 July 2024).

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
