# Peer review of "The Six-Transmembrane Enzyme GDE2 Is Required for the Release of Molecularly Distinct Small Extracellular Vesicles from Neurons"

_cells, 2024, doi:10.3390/cells13171414_

Round 1

Reviewer 1 Report

Comments and Suggestions for Authors

This is a relevant work in extracellular vesicles (EVs). It shows that Glycerophosphodiester Phosphodiesterase 2 (GDE2 or GDPD5), a six-transmembrane 12 protein that cleaves the Glycosylphosphatidylinositol (GPI)-anchor from proteins, controls EV release in Hek cells and neurons.

The authors overexpress GDE2 in Hek293T cells and show an increase in EV release, and when they purify EVs from GDE2, knockout neurons reveal a decrease in EVs. One primary concern relates to how data have been normalized. The total number of particles is usually normalized on the total number of cells or the total protein amount derived from the lysis of releasing cells.

A similar request for immunoblotting. Can the authors normalize the specific signals from the different antibodies on the total proteins on the membrane or a loading control like flotillin-1?

Most experiments have been performed in Hek293T cells, but the title refers to neurons. How neuronal EVs are similar to Hek293T cells EVs? The authors should consider changing the title.

No functional experiments have been performed with the GDE2 EVs. The authors claim EVs have synaptic and anti-oxidant functions, so viability experiments or oxidative stress markers should be evaluated.

Minor

The proteomics results take time to read. The paragraph should be re-framed to convey the key messages.

Author Response

RESPONSE TO REVIEWS: We thank the reviewers for their comments, which were positive and enthusiastic about the manuscript's relevance and interest to the EV community. Nevertheless, the reviews had several concerns. I detail below our responses to each reviewer. All changes are highlighted in the revised manuscript.

Reviewer #1:

The authors overexpress GDE2 in Hek293T cells and show an increase in EV release, and when they purify EVs from GDE2, knockout neurons reveal a decrease in EVs. One primary concern relates to how data have been normalized. The total number of particles is usually normalized on the total number of cells or the total protein amount derived from the lysis of releasing cells.

A similar request for immunoblotting. Can the authors normalize the specific signals from the different antibodies on the total proteins on the membrane or a loading control like flotillin-1?

  • We thank the reviewer for pointing this out. We have now clarified in the Materials and Methods Section of the revised manuscript how data were normalized for the nanoflow experiments and the immunoblotting experiments.

For the nanoflow studies, the same number of cells were used in each experiment (we have added absolute numbers in the text), with no observable differences in viability or health between each experimental condition (eg between transfections or between WT and Gde2KO neurons). For immunoblotting experiments, we utilized the same number of cells for each experimental group and detected no visible differences in viability or morphology between the groups. The amount of protein in EVs was normalized to the amount of protein in the lysate to provide a measure of the amount of released protein relative to the amount of total protein expressed.

Most experiments have been performed in Hek293T cells, but the title refers to neurons. How neuronal EVs are similar to Hek293T cells EVs? The authors should consider changing the title.

  • The heterologous HEK293T system was used to explore potential roles for GDE2 in EV release before moving to primary neurons, the major site of GDE2 expression in the nervous system. Here we utilized Gde2KO neurons to demonstrate a physiological requirement for GDE2 in releasing neuronal EVs and further utilized proteomics to identify possible functions for EVs released by GDE2 in neurons. Given that we have shown a physiological requirement for GDE2 in releasing EVs from neurons, it seems appropriate to keep the title as is.

No functional experiments have been performed with the GDE2 EVs. The authors claim EVs have synaptic and anti-oxidant functions, so viability experiments or oxidative stress markers should be evaluated.

  • The functional experiments suggested by the reviewer are not trivial and would take several months or even a year to develop and perform and are thus outside of the scope of this study. The timeframe for the suggested experiments would also not align with the 10-day turnaround time requested by the editors.

Minor

The proteomics results take time to read. The paragraph should be re-framed to convey the key messages.

  • Thank you for this feedback. We have modified the paragraph to improve clarity by removing the text explicitly restating Fold Change (FC) and False Discovery Rates (FDR), referring readers instead to the Supplemental Tables where this information is located. We have also restructured some sentences. We believe these changes decrease repetition and improve clarity.

Reviewer 2 Report

Comments and Suggestions for Authors

The manuscript „The six-transmembrane enzyme GDE2 is required for the release of molecularly distinct ectosomes from neurons“ from Shuler and collegues used gain- and loss-of-function approaches to investigate the role of GDE2, a GPI-anchor cleavage enzyme, on EV secretion quantitatively as well as qualitatively.

Many papers on EV research focus on methodology or clinical settings and I welcome that this paper investigates the role of a lipid modifiying enzyme as a modulator of EV content. The authors used OE of GDE2 WT and a activity-dead mutant in HEK293 cells as well as mouse neuronal WT and GDE2-KO cells to look at EV release.

The activity dead enzym is a good proof of principle experiment and it would be good to see a similar rescue experiment with GDE2 KO cells, as the effect of GDE2-KO is not very strong and the effect on EV markers not significant. 

Some methodological questions remain in my view, and the new MISEV guidlines should be followed (and cited) regarding naming of respective vesicles and markers as well as controls: Minimal information for studies of extracellular vesicles (MISEV2023): From basic to advanced approaches, Welsh et al., Journal of extracellular vesicles 13 (2), e12404

For example, please indicate in the M&M section absolute numbers of cells used for the different experiments as well as biological replicates.

EV namerd after their size rather then from their origin - large EV (formerly known as ectosomes or microvesicles) and sEV (exosomes). How can we be sure the GDE2-dependent EV are coming from the surface?

Question regarding the GDE2 overexpression which increases EV marker, EV numbers and protein loading. How was it normalized, was there a BCA difference in these samples? What about viability of these cells in OE and KO? Other (morphological) defects of GDE2 OE or KO?

Question regarding the proteomics approach, which of these identified proteins have a GPI anchor? What percentage of the GDE2 -dependent EV cargo does have a GPI anchor? This could be an additional digramme.

The passages about the proteomic analyses could benefit from a slightly better structure, as it is hard to follow through the text.

Taken together, an interesting article that can be easily improved, which is of interest to the EV and lipid enzyme community.

Author Response

Reviewer #2:

The activity dead enzym is a good proof of principle experiment and it would be good to see a similar rescue experiment with GDE2 KO cells, as the effect of GDE2-KO is not very strong and the effect on EV markers not significant.

  • Transfecting primary neurons is not trivial and would require the generation of lentivirus or adeno-associated virus (AAV) expressing GDE2 and GDE2H243.A, a process that can take several months. Further, the generation of neuronal cultures can also take several weeks (3 weeks to generate pups and 14 days growth in culture). Thus, the suggested experiment by the reviewer is not within the scope of the study and the 10-day timeframe requested by the editors.

Some methodological questions remain in my view, and the new MISEV guidlines should be followed (and cited) regarding naming of respective vesicles and markers as well as controls: Minimal information for studies of extracellular vesicles (MISEV2023): From basic to advanced approaches, Welsh et al., Journal of extracellular vesicles 13 (2), e12404

  • We thank the reviewer for this valuable input. We have made the requested changes according to the MISEV guidelines.

For example, please indicate in the M&M section absolute numbers of cells used for the different experiments as well as biological replicates.

  • We have inserted the required information in the text as suggested by the reviewer.

EV namerd after their size rather then from their origin - large EV (formerly known as ectosomes or microvesicles) and sEV (exosomes). How can we be sure the GDE2-dependent EV are coming from the surface?

  • Thank you for the question. Our studies show that EVs released by GDE2 contain known markers of EVs originating from the cell surface, specifically Annexin A1, CD9, and BSG (Matthieu et al., 2021). No changes were observed in known exosomal proteins such as Alix. Further, EVs released by GDE2 are dependent upon GDE2 GPI-anchor cleavage function, which is established to occur only at the cell surface (Salgado-Polo et al., 2020). These observations suggest that the majority of EVs released by GDE2 originate from the cell surface. We have modified the text throughout the manuscript to convey that the EVs released by GDE2 are likely (rather than definitively) to be ectosomes.

Question regarding the GDE2 overexpression which increases EV marker, EV numbers and protein loading. How was it normalized, was there a BCA difference in these samples? What about viability of these cells in OE and KO? Other (morphological) defects of GDE2 OE or KO?

  • We have now included more details in the Methods section of the revised manuscript that address these points. Specifically, for immunoblotting experiments, we utilized the same number of cells for each experimental group and detected no visible differences in viability or morphology between the groups for OE. The amount of protein in EVs was normalized to the amount of protein in the lysate to provide a measure of the amount of released protein relative to the amount of total protein expressed. There were no obvious changes in viability or morphology between the WT and Gde2KO neurons in culture.

Question regarding the proteomics approach, which of these identified proteins have a GPI anchor? What percentage of the GDE2 -dependent EV cargo does have a GPI anchor? This could be an additional digramme.

  • Thank you for raising this point. We detected only a couple of GPI-anchored proteins in the EVs. We have thus included their mention in the discussion rather than including a separate diagram.

The passages about the proteomic analyses could benefit from a slightly better structure, as it is hard to follow through the text.

  • Thank you for this feedback. We have revised these sections to make them clearer. Specifically, we have modified the passages by removing the text explicitly restating Fold Change (FC) and False Discovery Rates (FDR) and referring readers instead to the Supplemental Tables where this information is located. We have also restructured some sentences. We believe these changes reduce repetition and improve clarity significantly.

Round 2

Reviewer 1 Report

Comments and Suggestions for Authors

The authors made appropriate changes, taking into consideration some of my questions. I suggest again downplaying the tone on the fact that EVs have synaptic and anti-oxidant functions because no functional experiments have been performed. 

Author Response

We thank the reviewer for their comments to tone down the idea that EVs have synaptic and anti-oxidant functions. We have made changes throughout the manuscript explicitly stating that these proposed functions require further validation and functional evaluation. These changes are highlighted in blue. We have also highlighted in green statements in the original manuscript where we stated that these functions remain speculative.

In response to the editor's suggestion that we include more comments about controls in reference to Reviewer 2, we included text in the Methods section of the previously revised Manuscript that clarifies the controls for the immunoblotting experiments. Please note that we also included text in the Methods section that included clarification of the controls for the nanoflow experiments, where the same number of cells were used in each experiment, with no observable differences in viability or health between each experimental condition (eg between transfections or between WT and Gde2KO neurons). We also included the absolute numbers of cells used for each experiment in the Methods section, as well as the number of biological replicates used for each study. We have now included additional details regarding controls in the revised text for further clarification-these are highlighted in blue.
